# Cognitive Functioning in Children and Young People with Congenital Heart Disease: A Systematic Review of Meta-Analyses

**DOI:** 10.3390/healthcare12242594

**Published:** 2024-12-23

**Authors:** Maria Koushiou, Sauleha Manzoor, Antonis Jossif, Nuno Ferreira

**Affiliations:** 1Department of Social Sciences, School of Humanities and Social Sciences, University of Nicosia, Nicosia 2417, Cyprus; koushiou.m@unic.ac.cy (M.K.); manzoor.s@live.unic.ac.cy (S.M.); 2Paedi Center for Specialized Pediatrics, Athalassa Avenue, 178, Nicosia 2025, Cyprus; ajossif@paedi.org.cy

**Keywords:** congenital heart disease, cognitive functioning, narrative synthesis, neurodevelopmental outcomes

## Abstract

**Objectives**: Congenital heart disease (CHD) is a prevalent condition affecting young people that often necessitates complex medical interventions. This study aims to provide a synthesis of existing meta-analyses examining the impact of CHD on the cognitive functioning in children and young people; **Methods**: A comprehensive literature review was conducted, identifying peer-reviewed meta-analyses from 1 January 1976 to 17 December 2024, focusing on the cognitive outcomes of CHD patients aged 0–24 years. Data extraction covered study details, participant demographics, outcomes, and quality assessments. Quality assessment followed the Scottish Intercollegiate Guidelines Network (SIGN) checklist, and a narrative synthesis was conducted; **Results**: The narrative synthesis highlighted significant cognitive impairments in children with CHD across domains such as Intelligence Quotient (IQ), executive functions (EF), memory, and alertness. Cognitive impairments were also observed to become worse with increasing age. Furthermore, surgical interventions were found to impact cognitive outcomes, with surgeries at an early age improving survival rates but not entirely mitigating cognitive deficits. Cognitive impairments were more marked in young people assessed at an older age and with a more complex CHD presentation. **Conclusions**: Children with CHD face persistent cognitive challenges, underscoring the need for early identification and tailored interventions. Routine cognitive assessments and comprehensive care, including educational and psychological support, are crucial for improving neurodevelopmental outcomes. Future research should focus on longitudinal studies to track cognitive development and explore effective rehabilitation programs.

## 1. Introduction

Congenital heart disease (CHD) refers to a type of birth defect that involves the malformation of heart walls, valves, and blood vessels, which affects the functioning of the heart [1]. This condition is present at birth and can vary in severity from simple defects to complex life-threatening complications [2]. CHD has been identified as an area of concern within pediatric healthcare due to its possible impact on neurocognitive functioning among children and young people [3]. Treatment for CHD depends on the type and severity and may include prescribing medications, undergoing catheter procedures, surgery, and heart transplants [4]. Advances in medical and surgical care have substantially improved survival rates for children with CHD, leading to a growing population of children and young people living with this condition [5]. However, alongside these improvements, there has been an increasing recognition of the cognitive and developmental challenges observed among this population.

Neurodevelopmental implications associated with CHD are particularly significant. Research suggests that factors such as reduced cerebral blood flow, chronic hypoxia, and the effects of surgeries can adversely affect brain development, leading to various deficits in cognitive functioning [6,7,8]. Cognitive functioning encompasses various mental processes such as attention, executive functions, memory, language, and problem-solving abilities. Attention is crucial for focusing on specific stimuli or tasks while filtering out distractions, playing a key role in learning and daily activities [9]. Memory involves the processes of encoding, storing, and retrieving information and can be categorized into short-term memory (temporary storage of information), working memory (manipulating information for cognitive tasks), and long-term memory (permanent storage of knowledge and experiences) [10]. Additionally, cognitive ability is often assessed using the Intelligence Quotient (IQ), which summarizes performance on standardized tests and provides insights into an individual’s cognitive capacity relative to the general population [11]. The Full-Scale IQ (FSIQ) is a comprehensive measure of a person’s intellectual ability, that combines composite scores obtained from verbal and non-verbal tasks. Among various cognitive processes, executive functioning (EF) is particularly significant, encompassing skills such as planning, impulse control, problem-solving, and strategic and goal-directed behavior [12,13]. EF and IQ rapidly develop in children and are both essential in carrying out basic and complex tasks successfully and fostering academic achievement [14,15].

### Purpose of the Present Study

Existing research and healthcare initiatives have mainly focused on establishing essential medical and surgical interventions for the survival of children with CHD. As the number of individuals with CHD continues to grow, it becomes increasingly important to acknowledge and address the cognitive effects of the condition on children and young individuals. Despite the expanding research on cognitive outcomes in children with CHD, there are significant gaps in the literature. While individual studies offer valuable insights, a systematic review can coherently aggregate and interpret these findings, considering the variability in study designs, populations, and outcome measures. Therefore, in this paper, we systematically analyze meta-analyses that focus on the cognitive functioning of children and young people with CHD, offering a comprehensive narrative synthesis of the existing literature. Additionally, we also explore potential areas for practical implications and consider directions for future research.

## 2. Materials and Methods

### 2.1. Information Sources and Search Strategy

Meta-analyses that examined cognitive functioning in children and young people with CHD published from 1976 until 27 November 2024 were considered. The literature search was conducted using PubMed, PsycINFO, Science Direct, EMBASE, Google Scholar, and Cochrane electronic databases with the following search terms: (cognitive function OR cognitive outcomes OR neurodevelopmental outcomes) AND (congenital heart disease) AND (children OR adolescents) AND (meta-analysis). The search was revised post-review for EMBASE and Science Direct on 17 December 2024. In this paper, the term ‘young people’ has been defined as individuals aged 10–24 years, consistent with the World Health Organization’s (WHO) categorization in their reports on adolescent health [16]. A systematic review of these articles was conducted, and the systematic review protocol was registered on Prospero (Registration nu.: CRD42023461735).

### 2.2. Study Selection

Studies were included if they met the following eligibility criteria: (1) were observational, peer-reviewed meta-analyses that clearly stated neurodevelopmental outcomes; (2) reported quantitative results (e.g., scores based on standardized tests/questionnaires for measuring neurodevelopmental outcomes) in children and/or young people (under the age of 25 years) with CHD compared to a reference/control group; (3) were published in the English language after 1 January 1976, as CHD was only defined as a diagnostic category in 1976 [17,18]; or (4) the study examined at least one neurodevelopmental cognitive outcome.

Studies were excluded if they (1) focused solely on adult samples; (2) were questionnaire-based studies exploring quality of life; (3) were intervention studies, qualitative studies, or case reports; (4) were restricted to specific patient sub-groups (e.g., preterm births, patients with Down syndrome or autism, heart transplant recipients, etc.); (5) included participants with multi-organ syndromes, which could act as a confounder; (6) were dissertation, book, book chapter, or review-type articles; or (7) studies outside the specified age range were included only if the authors provided separate data for children and young adults (under the age of 25 years).

The PRISMA 2020 [19] flow diagram, seen in Figure 1, was utilized to document the systematic search across various databases mentioned above to identify and select articles for inclusion in this study. A total of 1631 records were identified through searches across PubMed (11), PsycINFO (80), Science Direct (988), EMBASE (157), Cochrane (10), and Google Scholar (385, after exploring search results up to page 15 of approximately 19,400 results). After the removal of duplicates, 1403 unique records were retained for screening. All records identified were imported into the Mendeley Reference Manager 1.19.5 software, a platform which facilitated the identification and removal of duplicate records. The screening of all titles and abstracts to identify papers for full-text review was performed by the author S.M., while the studies retrieved from post-initial screening were reviewed independently by the other authors (M.K. and N.F.). Krippendorf’s alpha coefficient [20] for nominal data was calculated to assure a high level of agreement between raters. All points of disagreement were resolved by all authors collaboratively. For the detailed search strategy, see Appendix A.

### 2.3. Methodological Quality

The quality of the eligible papers was evaluated using the Scottish Intercollegiate Guidelines Network (SIGN) checklist for systematic reviews and meta-analyses [21]. The methodological quality of each study was rated as follows: high quality (majority of criteria met, with little or no risk of bias), acceptable (most criteria met with some flaws in the study and/or with some risk of bias), low quality (most criteria not met, or significant flaws relating to key aspects of study design), and reject (poor quality with significant flaws). Author S.M. reviewed all articles meeting eligibility criteria and assigned half of the articles at random to author M.K. and author N.F. for independent screening to mitigate selection errors and bias. Each study’s quality rating was determined based on the number of criteria that it fulfilled from the SIGN checklist, with particular attention to the risk of bias in the study design, data collection, and reporting methods. If a study met most of the SIGN criteria with only minor issues, it was rated as high quality; studies with some flaws that might influence the results were rated as acceptable; those with significant flaws were rated as low quality; and those with major issues were rejected. In cases where discrepancies in quality ratings occurred, the authors compared their assessments and engaged in discussions to reach a consensus, ensuring a more reliable evaluation of each paper’s methodological quality.

### 2.4. Data Extraction

Information from the included studies was extracted by the first author and then reviewed by the authors M.K. and N.F, who followed a predefined data extraction protocol to ensure consistency. The following data were extracted, as shown in Table 1: authors, year of publication, number of studies considered for meta-analysis, age range, heterogeneity, quality assessment tools used in the study, and the SIGN quality assessment outcome. This was then used to produce short textual summaries for the systemic narrative analysis.

## 3. Results

The narrative synthesis reviews and summarizes findings from the selected studies, outlining the cognitive challenges linked to CHD, the impact of surgical treatments, and the probable ramifications for those affected. Research studies seem to report, across the board, that children with CHD display lower general cognitive functioning compared to their healthy peers. Furthermore, the influence of surgical interventions appears to be crucial in shaping cognitive development and impacting the overall growth of pediatric patients with CHD.

### 3.1. Intelligence Quotient (IQ)

Across studies, as seen in Table 2, children with CHD consistently exhibited lower IQ scores compared to their healthy counterparts. Sterken et al. (2015) [27] observed deficits notably within the domain of intelligence and reported a medium standardized mean difference (SMD) for intelligence at −0.53 (95% CI −0.68–−0.38), indicating significant differences compared to healthy controls. Sistino et al. (2012) [25] reported that, despite advancements in the past two decades, the average IQ and developmental scores remained suboptimal, with an overall mean IQ of 85.9 (95% CI 82.3–89.5) in children with Hypoplastic Left Heart Syndrome (HLHS) undergoing S1N. Siciliano et al. (2019) [24], who also examined children with HLHS, found significant deficits in Full-Scale IQ (FSIQ) and Performance IQ (PIQ), with moderate effects on Verbal IQ (VIQ) as compared to normative data. Feldman et al. (2021) [22] noted a decrease of 9.9 IQ points in children with congenital heart disease (CHD) compared to healthy controls, with significant heterogeneity in the data (Total IQ I^2^ = 94.7%). Soares et al. (2023) [28] found that, among children aged 4 to 5 years with the transposition of the great arteries surgically corrected during the neonatal period, the mean global FSIQ was 97.5 (95% CI 90.0–104.9), with severe heterogeneity (I^2^ = 94%, *p* < 0.01). Their mean PIQ was 92.9 (95% CI 89.7–96.2), and their mean VIQ was 95.1 (95% CI 93.0–97.2), indicating moderate deficits. Furthermore, 22.3% of children in this group scored less than one standard deviation below the mean, a proportion not significantly different from the general population.

Additionally, Karsdrop et al. (2007) [23] showed that patients with severe CHD exhibited significantly lower Verbal (VIQ r(20) = −0.62, *p* = 0.003) and Performance Intelligence (PIQ r(20) = −0.59, *p* = 0.006) compared to normative data, highlighting a notable decline in cognitive functioning and a persistent trend in cognitive deficits across various age categories. Feldman et al. (2021) [22] observed that older age at the time of evaluation was associated with lower IQ scores, pointing to the fluctuating nature of cognitive deficits in CHD populations. Siciliano et al. (2019) [24] emphasized the increasing cognitive deficits in children with CHD as they age, highlighting the importance of early interventions, with meta-regression analyses showing a decrease of 1.1 FSIQ points for each year increase in average participant age, stressing a potential progressive cognitive decline among children and young adults with CHD.

Furthermore, Snookes et al. (2010) [26] highlighted the presence of developmental delays in cognitive domains such as language, attention, and problem-solving skills, evident in children who underwent cardiac surgery before the age of 6 months. In accordance with these findings, Soares et al. (2023) reported that a higher proportion of children with a surgically corrected transposition of the great arteries score one standard deviation below the mean on mental and psychomotor developmental indices as well as on motor and language composite scores from one to three years of age, as compared to the general population. Sterken et al. (2015) [27], who also investigated the post-operative neurocognitive performance of pediatric patients with CHD, showed that individuals with CHD displayed poorer performance in neurocognitive domains such as intelligence, executive function, attention, and memory when compared to their healthy counterparts. Sistino et al. (2012) [25] linked increased survival rates from surgical interventions to improved but still below-normal neurodevelopmental outcomes.

### 3.2. Executive Functioning (EF)

EF was another domain where children with CHD displayed notable impairments. Sterken et al. (2015) [27] noted significant impairments in EF and, specifically, in the inhibition function with moderate SMD and high heterogeneity when comparing children with CHD (with a median age of 7.35 years at testing) to their healthy peers. Feldman et al. (2021) [22] observed that EF deficits were consistent, with medium effect sizes observed for performance-based EF (referring to structured tasks or tests designed to measure cognitive processes such as planning, flexibility, and inhibitory control) in children aged 5 to 17 years (and a median age of 9.76 years) compared to healthy controls.

### 3.3. Memory and Attention

Sterken et al. (2015) [27] observed mild impairment (small SMD) in verbal working memory and immediate memory among individuals with CHD under 24 years of age compared to healthy controls, while no significant differences were noted on non-verbal memory between the two groups. Also, deficits were observed regarding the level of alertness (alertness non-reaction time and alertness reaction time), an attentional function characterized by the ability to maintain a state of responsiveness and readiness to react to stimuli exhibited by individuals with CHD when compared to healthy controls [9,27]. Deficits in both memory and alertness in children with CHD can impact their ability to perform on IQ tests and tasks requiring complex EF skills.

### 3.4. Overview

The presence of cognitive issues among individuals with CHD included difficulties in intelligence and executive functions, with predominant difficulties in the inhibition function, attention, and in reaching overall developmental milestones, such as motor skills [29]. With effect sizes ranging from medium to large, five out of the seven selected studies noted impairments on intelligence tests that particularly affected the composite scores of total IQ and Performance IQ in children and youth with CHD as compared to the normative mean or healthy controls. CHD subtype had a modifying effect on intelligence with more severe CHD (HLHS and UVH) associated with lower scores on total and Performance IQ as compared to milder CHD subtypes (ASD/VSD) [22,23]. A modifying effect of age was evident in some studies but not all. For example, Karsdorp and colleagues (2007) [23] noted that cognitive difficulties remained relatively stable across different age groups whereas, in Siciliano and colleagues (2019) [24] and Feldman and colleagues (2021) [22], greater sample age was associated with lower total IQ scores. Overall, regarding the performance of children with CHD on intelligence tests, Sistino and colleagues (2012) [25] noted a significant increase of mean IQ from 1989 to 1999 possibly reflecting advances in surgical procedures implemented with children with HLHS. In addition, it is generally acknowledged and evident in the summative data shared in Table 2 that children with CHD perform within the normal or low-normal range in IQ tests with a reduction of a few points (e.g., 9.9 points out of the normative mean of 100 in Feldman and colleagues, 2021 [22]). Although only present in a few points, this discrepancy from the normative mean might have important implications in the individual’s educational and later professional life and their families.

Surgical intervention effects were also assessed in relation to children and youth performance in cognitive tasks. Children in middle childhood with HLHS who underwent S1N showed improvements on their cognitive outcomes, but their scores on IQ tests and Bayley developmental tasks were still below normal [25]. In addition, children who received cardiac surgery during their infancy (under 6 months of age) presented cognitive and motor difficulties in early childhood (from one to three years of age) with more pronounced motor risks around the age of one year [26]. Cardiac surgery in the neonatal period in children with a transposition of the great arteries was also found to impact cognitive, motor, and language development in children from one to three years of age, as they performed at the low end of the normative reference range with significant heterogeneity in most these meta-analytical results [28]. In addition, worse scores on intelligence tests, EF, and, especially, inhibition and attention (alertness) were evident in youth under the age of 24 who underwent heart surgery or another interventional cardiac procedure [27].

In summary, the extent of cognitive impairment varied across studies, influenced by factors such as the severity of CHD, its subtype, the type of treatment received, and the age at which individuals were evaluated. These variations seem to impact the interplay of medical, developmental, and environmental elements that shape cognitive outcomes for those with CHD. While there are shared cognitive challenges among CHD patients, discrepancies in timing, severity, and affected domains emphasize the need for personalized intervention strategies tailored to each patient’s unique circumstances.

## 4. Discussion

The findings of this systematic analysis of meta-analyses provide insight into the cognitive challenges that young individuals with CHD encounter. The reviewed meta-analysis studies indicate that significant cognitive impairments in a number of domains, such as IQ, EF, attention, and memory, are associated with CHD. Children with CHD continue to show lower cognitive functioning than their healthy peers, despite advances in medical and surgical therapies. This highlights the persistent nature of the effect of CHD on a person’s cognitive functioning [22,24,25,27].

The potential underlying mechanisms of impaired cognitive functioning include early brain injury, fewer opportunities for social engagement during early childhood, and the stress and anxiety driven by long-term medical conditions [30]. More specifically, an increasing number of studies point to the role of altered cerebral blood flow with impaired cerebral oxygenation both in utero and after birth in children with CHD affecting their brain development. It is, thus, indicated that their brain is less mature at birth than suggested by gestational age, which subsequently affects brain growth and maturation during the neonatal period [31,32,33]. Brain growth and maturation, myelination, and neuronal development occurring during these critical periods in a child’s life are compromised due to CHD effects (for further discussion, see [34]). These effects are further exacerbated in cases of CHD with comorbidities, such as prematurity, and/or genetic abnormalities or syndromes [29].

The evidence from the narrative synthesis suggests that there is an effect of CHD on cognitive outcomes, with CHD contributing to neurodevelopmental deficits. In a thorough investigation [22], cognitive outcomes in school-age children (ages 5 to 17 years) with CHD showed that, although the participants’ overall intellectual capacity fell within the normal range, it was marginally lower by 9.9 IQ points compared to their healthy counterparts. Significantly, EF appeared to be compromised, indicating a notable decline in skills such as cognitive flexibility, working memory, and inhibition [22]. These findings are consistent with another review study [27] that also observed considerable impairments in EF (inhibition function, attentional alertness, and verbal memory) following surgical interventions among children and young people with CHD under 24 years of age. The variability in effect sizes and heterogeneity across studies [22] indicates that, while there is a general trend of cognitive deficits in CHD populations, individual outcomes can vary widely. This variability may be attributed to differences in the complexity of CHD, the timing and type of surgical interventions, and the age at which cognitive assessments were conducted.

Moreover, the complexity of CHD and the age of assessment seem to be significant determinants of cognitive functioning. Children with more severe forms of CHD (e.g., HLHS, UVA, TGA) tend to exhibit greater cognitive deficits [22,23] as compared to milder CHD (ASD/VSD), and the impact of CHD on cognitive outcomes appears to intensify with age [22,24]. Meta-regression analyses indicate a progressive decline in cognitive scores with increasing age, emphasizing the need for early identification and intervention [24]. However, this finding stems primarily from observational data and, therefore, should be taken with a degree of criticism. These findings underscore the importance of continuous monitoring and tailored educational and therapeutic interventions to support cognitive development in this population throughout their development.

Subsequently, among children and young individuals, academic or school performance is an area that is considerably impacted by CHD. In a recent systematic review and meta-analysis [35] focusing on the academic outcomes and special education needs of children with congenital anomalies, children with severe CHD had significantly higher odds of requiring special education services and tended to underachieve academically. Factors such as the severity of the congenital anomaly and socioeconomic deprivation were identified as significant predictors of poor academic outcomes. These findings align with the results of this review, which identified cognitive impairments, particularly in IQ and EF, as common challenges faced by children with CHD [22,24]. Impairments in EF—including difficulties with problem-solving and cognitive flexibility—could exacerbate challenges within the classroom, further reinforcing the link between neurodevelopmental outcomes and academic performance [25,27]. The link between these cognitive deficiencies and academic challenges underscores the importance of early cognitive screening and the development of tailored educational support to help children with CHD succeed academically.

Moreover, the impact of surgical intervention seems to play a pivotal role in influencing cognitive progress and affecting the overall development of children and young people with CHD. Early heart surgery seems to have a negative impact on motor and cognitive abilities in early childhood, as assessed via the Bayley Scales of Infant Development (mental development index and the psychomotor developmental index) [28] and with profound risks for motor skills reported around the age of one year old [26]. This suggests that, although early surgeries may alleviate certain cognitive deficiencies, they do not completely avert motor impairments, which can impede overall growth. Enhancements in survival rates following the Stage I Norwood (S1N) procedure for HLHS were linked with better yet subpar neurodevelopmental results in children during middle childhood [25]. More specifically, despite advancements in surgical methodologies, average IQ and developmental scores remained at the low end of the normative reference range for preschoolers with a surgically corrected transposition of the great arteries [28], while, in children aged 1 to 12 years, intelligence score persisted below average levels as compared to healthy controls [25]. Furthermore, a troubling rise in ADHD occurrences among S1N procedure recipients was also identified. These reports highlight the intricate balance between the possible effects of undergoing life-saving surgical interventions and the enduring cognitive and behavioral ramifications for CHD-afflicted young individuals. Although persistent cognitive deficits into adolescence and adulthood have been highlighted [23,24], these have stemmed primarily from cross-sectional studies, underscoring the need for the longitudinal monitoring of these outcomes.

More difficult and frequent surgeries have been associated with acquired cognitive impairments in children with severe CHD. These cognitive impairments may reflect brain disturbances. For example, before and after congenital heart surgery, 20% of infants had preoperative white matter injury, and an additional 42% had new white matter injury in the postoperative period [36]. Disturbances in white matter microstructure have been found to contribute to cognitive compromise in adolescents who underwent open-heart surgery in infancy [37]. Similarly, reduced brain volumes and cortical thickness were documented in children aged 10–19 years old with CHD post-Fontan surgery, as compared to controls [38]. Even though advances in surgical practices (e.g., the use of modified Blalock–Taussig shunt as compared to the right ventricular to pulmonary artery RVPA conduit) have significantly contributed to improvements in neurodevelopmental outcomes [25], other evidence suggests that patient and preoperative risk factors (such as low birth weight or gestational age and social class), as well as cumulative postoperative morbidity (e.g., longer postoperative mechanical ventilation or hospital stay), may have greater contributions to outcomes than specific operative techniques [39].

Furthermore, the impact of CHD on the psychological and social aspects is profound and should not be ignored. Even though this is not within the scope of the current review, it is important to highlight that many young individuals with CHD encounter behavioral issues and social obstacles added to their cognitive difficulties. Karsdrop et al. [23] highlighted the need for psychological interventions to manage behavior and mental health issues, such as anxiety and depression, which are observed to be more prevalent among older children and adolescents with CHD. In a study exploring how CHD affects the mental and social well-being of children and adolescents [40], about 25% of participants reported facing behavioral difficulties, surpassing the rates observed among healthy counterparts. The persistence of these challenges across different age brackets suggests the importance of consistent early interventions to address both cognitive and psychosocial obstacles.

In conclusion, there is an increasing number of empirical studies on the cognitive and neurodevelopmental outcomes of children and youth with CHD. However, there is great heterogeneity between published studies in terms of population recruited, samples’ age, CHD subtype, medical interventions involved, and assessment tools used. Due to this heterogeneity, aggregated results are needed to determine the cognitive domains that are more affected in CHD populations and shed light on the factors affecting performance in these domains. The present systematic review responds to this gap in the literature by providing a narrative summary of meta-analysis data. These findings could be used to enrich or even refine existing screening and assessment programs by determining subpopulations with CHD that are at higher risk for specific cognitive impairments and by suggesting how medical interventions and age can influence risk in these populations. In addition, the present study provides insight as to the empirical quality of existing studies and gives an overview of their limitations, which can be further used to guide and inspire future research.

### 4.1. Implications for Practice

Practical implications of these findings include the necessity for routine cognitive assessments for children with CHD to identify early deficits and implement timely interventions. In particular, early childhood, ages 0 to 5 years, is a critical period for brain development, during which interventions can have the most substantial impact [29]. Early screening should focus on key developmental domains identified in this review, including IQ, executive function (EF), inhibition, attentional alertness, and verbal memory, as these areas are frequently compromised in children with CHD [22,24]. Implementing personalized educational and therapeutic strategies tailored to each individual’s unique needs can help address specific cognitive deficits and support overall development. Additionally, integrating interdisciplinary approaches to improve neurodevelopmental outcomes should be considered. Cardiology centers tend to have limited access to child and adolescent psychiatrists, psychologists, and behavioral pediatricians [41]. Collaboration with these specialists can provide comprehensive care that addresses cognitive, behavioral, and emotional challenges. These specialists can contribute to the development of procedural preparation programs and provide tools, ensuring that they effectively meet the needs of those with CHD. A practical example of this type of collaboration can be found in the Cardiac Neurodevelopment Outcomes Collaborative (CNOC; https://cardiacneuro.org/; accessed 5 December 2024) initiative that brings together more than 52 hospitals and institutions dedicated to the study and care of neurodevelopmental impact in CHD.

In addition, healthcare providers should be aware of the potential long-term cognitive implications of CHD and surgical interventions, advocating for comprehensive care approaches that include neurodevelopmental screening, surveillance, and assessment following good practices and guidelines [29]. Educational and psychological support services should be integrated into the care plans for children and young people with CHD to address specific cognitive challenges and enhance their academic, social, and psychological outcomes.

### 4.2. Strengths and Limitations of the Review

The strengths of this systematic review include the rigorous methodology following a clear and consistent criterion for the inclusion and exclusion of studies, adherence to PRISMA guidelines, and comprehensive search strategy. However, limitations include sources of heterogeneity such as variations in cognitive assessment tools, lack of longitudinal data, cohort studies, studies repeated across the different meta-analyses, and potential publication bias. Also, data from non-European populations and developing countries were limited, and studies often do not include Indigenous and tribal populations. Reported heterogeneity across the different meta-analyses reviewed was high, except for the Intelligence, alertness non-reaction time, alertness reaction time, verbal memory, and EF non-reaction time parameters in Sterken at al. [27]. The lack of sensitivity analyses in the studies examined here also affects the potential impact of the findings. Only one study conducted a sensitivity analysis [22] and this was limited to the exclusion of non-English language studies, which did not affect the main effects of the meta-analysis. Future meta-analyses on this topic should include more extended sensitivity analyses. Finally, increased survival rates of children with more severe forms of CHD can skew more recent studies, as they would increase the potential pool of participants with cognitive difficulties.

### 4.3. Research Implications and Recommendations

Future research could also focus on longitudinal studies to track cognitive development over time and identify critical periods for intervention. Investigating the underlying mechanisms of cognitive impairments in CHD or delving into its subtypes could provide insights into potential therapeutic targets. Also, future reviews could aim to standardize outcome measures and include grey literature to provide a more accurate estimate of cognitive deficits in CHD populations. Additionally, research should explore the effectiveness of specific cognitive rehabilitation programs and educational interventions tailored to the needs of children with CHD.

## 5. Conclusions

Overall, this systematic review highlights the significant cognitive challenges faced by children and young people with CHD. While there is strong evidence to suggest that CHD contributes to cognitive deficits, the extent and nature of these impairments vary. Surgical interventions, the complexity of CHD, and age are critical factors that influence cognitive outcomes. These findings underscore the importance of early and ongoing cognitive assessments and interventions to support the neurodevelopmental needs of this vulnerable population.

## Figures and Tables

**Figure 1 healthcare-12-02594-f001:**
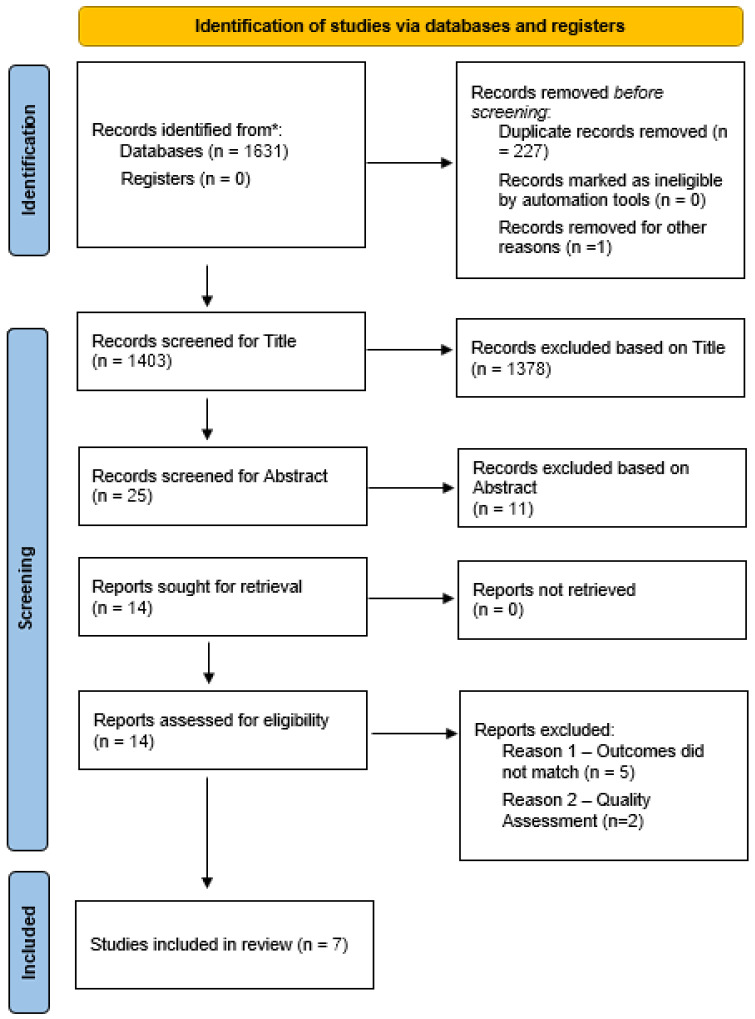
PRISMA flow diagram. Note: * Reporting the number of records identified from each database or register searched (rather than the total number across all databases/registers).

**Table 1 healthcare-12-02594-t001:** Summary of studies and quality assessment.

Authors (Year of Publication)	Number of Studies	Age Range	Heterogeneity	Quality Assessment Tools Used in the Study	SIGN Quality Assessment Outcome
Feldmann et al. (2021) [22]	IQ—74 (VIQ: 32, PIQ: 23), EF—13	5–17	Total IQ (I^2^ = 94.7%), VIQ (I^2^ = 96.5%), and PIQ (I^2^ = 79.6%) Behavior-rated EF (I^2^ = 51.3%) Performance-based EF (I^2^ = 64.5%)	SIGN and Newcastle–Ottawa Scale	Acceptable
Karsdorp et al. (2007) [23]	25	2–19	Reported heterogeneity in the dataset; I^2^ values were not specifically mentioned	-	Acceptable
Siciliano et al. (2019) [24]	13	2.6–17	Significant heterogeneity present in all models	NIH Quality Assessment Tool for Observational Cohort and Cross-Sectional Studies	High Quality
Sistino et al. (2012) [25]	14 for neurodevelopmental outcomes	1–12	Significant heterogeneity in all models	-	Low Quality
Snookes et al. (2010) [26]	Cognitive outcome (11 early development, 4 preschool age, and 7 school age); motor outcome (11 early development, 2 preschool age)	Early development (1 to 3 years), preschool (3–5 years), and school (5 to 17 years).	Early development: BSID-II MDI: I^2^ = −55.9, BSID-II PDI: I^2^ = 69.1	-	Low Quality
Sterken et al. (2015) [27]	12	<24	Intelligence (I^2^ = 32%), Alertness Non-reaction Time (I^2^ = 19%), Alertness Reaction Time (I^2^ = 0%), Verbal Memory (I^2^ = 0%), EF Reaction Time (I^2^ = 80%), and EF Non-Reaction Time (I^2^ = 0%)	-	Acceptable
Soares et al., 2023 [28]	17	<5	1 year of age: BSID-II: PDI (I^2^ = 87%, *p* < 0.01); MDI (I^2^ = 95%, *p* < 0.01). BSID-III: motor and cognitive composite score (I^2^ = 0%, *p* = 0.80). 2 years of age: BSID-II: PDI (I^2^ = 83%, *p* < 0.01); MDI (I^2^ = 76%, *p* = 0.02). BSID-III: motor composite score (I^2^ = 69%, *p* = 0.07); cognitive composite score (I^2^ = 91%, *p* < 0.01); language composite score (I^2^ = 35%, *p* = 0.21) 3 years of age: BSID-II: PDI (I^2^ = 46%, *p* = 0.16); MDI (I^2^ = 0%, *p* = 0.89). 4 to 5 years of age: WISC: FSIQ (I^2^ = 94%, *p* < 0.01); PIQ (I^2^ = 54%, *p* = 0.14); VIQ (I^2^ = 0%, *p* = 0.97)	Cochrane’s RoB 2 Tool for randomized control trials and Cochrane’s ROBINS-I Tool for non-randomized studies	High Quality

Note: The numbers in this section represent the number of studies included in the meta-analysis for each neurodevelopmental outcome. Acronyms: SIGN, Scottish Intercollegiate Guidelines Network; WISC: Wechsler Intelligence Scale for Children; IQ, Intelligence Quotient; FSIQ, Full-Scale Intelligence Qvuotient; VIQ, Verbal Intelligence Quotient; PIQ, Performance Intelligence Quotient; EF, executive function; BSID: Bayley Scales of Infant and Toddler Development; MDI, mental developmental index; PDI, psychomotor developmental index.

**Table 2 healthcare-12-02594-t002:** Summary of key findings in the selected studies.

Authors (Year of Publication)	Outcomes Measured	Comparator Groups	Primary Results	Effect Size
Feldmann et al. (2021) [22]	Cognitive function (IQ),VIQ, PIQ, and EF (performance-based and behavior-rated).	Comparisons between different CHD subtypes (HLHS vs. ASD/VSD cohorts), and between children with CHD and healthy controls	Compared to healthy controls, children with CHD showed a decrease of 9.9 IQ points. The severity of cognitive functioning was modified by CHD subtype, with total and performance IQ being lower in HLHS and UVH cohorts, compared with that of ASD/VSD subtypes. Older age at assessment was associated with lower IQ scores in cohorts with transposition of the great arteries, whereas improvement of performance IQ was found for UVH cohorts in more recent publications. In addition, performance-based EF was lower in CHD children as compared to healthy controls, without a specific EF profile (that is, in regards to age or EF domain).	Overall estimate for children with CHD: IQ: 96.03 (95% CI 94.91 to 97.14) VIQ: 95.5 (95% CI 92.14 to 98.86), PIQ 96.61 (95% CI 94.59 to 98.64) Behavior-rated EF: 51.31 (95% CI 49.07 to 53.55) Children with HLHS: IQ: 88.47 (95% CI 84.39 to 92.55) PIQ: 88.3 (95% CI 83.33 to 93.35) Children with UVH: IQ: 92.65 (95% CI 90.00 to 95.29) Children with ASD/VSD: IQ: 98.51 (95% CI 95.83 to 101.20) PIQ: 100.46 (95% CI 99.28 to 101.64) CHD vs. Healthy controls: Total IQ: SMD = −0.85 (95% CI −1.08 to −0.62) Performance-based EF: SMD = −0.56 (95% CI −0.65 to −0.46)
Karsdorp et al. (2007) [23]	Cognitive Functioning (IQ), VIQ, and PIQ	CHD vs. healthy control group	Patients with severe CHD (e.g., HLHS, TGA) exhibited lower cognitive functioning than those with less severe CHD (e.g., VSD, ASD), specifically with respect to Performance Intelligence. Moreover, decreased cognitive functioning remained relatively stable across different age groups. Patients with more severe CHD (e.g., HLHS) exhibited lower PIQ and VIQ than would be expected by normative data.	IQ r(46) = −0.45, *p* = 0.002 VIQ r(20) = −0.62, *p* = 0.003, PIQ r(20) = −0.59, *p* = 0.006
Siciliano et al. (2019) [24]	FSIQ, VIQ, and PIQ	Children with HLHS vs. normative data	Scores on measures of cognitive function in children and adolescents with HLHS were significantly lower than the normative mean across all domains. Greater sample mean age was associated with greater deficits in FSIQ. One year of increased mean sample age reflected a decrease of 1.1 IQ points across studies.	FSIQ: g = −0.87, 95% CI (−1.10, −0.65); mean FSIQ across studies: 86.88 ranging from 70.40 to 94.90 PIQ: g = −0.89, 95% Cl (−1.11, −0.68); mean PIQ across studies: 86.56, ranging from 78.00 to 94.50 VIQ: g = −0.61, 95% (−0.96, −0.50); mean VIQ across studies: 90.82, ranging from 81.32 to 98.90
Sistino et al. (2012) [25]	WISC IQ scores and the Bayley II MDI and PDI scores	Children with HLHS undergoing S1N vs. normative data	The mean IQ increased significantly (*p* < 0.05) from 1989 to 1999. The mean MDI and PDI increased significantly (*p* < 0.05) from 1998 to 2005. Despite these significant improvements over the past two decades, the mean IQ and the Bayley II MDI and PDI are still below normal.	Wechsler IQ from 1989 to 1999: Overall mean: 85.09 95% CI (82.3–89.5) Bayley II MDI from 1998 to 2005: Overall mean: 86.9 (95% CI, 84.9–88.9) Bayley II PDI from 1998 to 2005: Overall mean: 73.4 (95% CI, 71.2–75.5)
Snookes et al. (2010) [26]	Cognitive (BSID-I, BSID-II MDI, Griffiths MDS, WPPSI-R, WISC-III), Motor (BSID-I, BSID-II PDI, PDMS)	Children who received cardiac surgery before the age of 6 months vs. normative data	In infants receiving cardiac surgery at under 6 months of age, cognitive and motor developmental domains were below the expected mean at all ages studied. It was reported that, at around 1 year of age, the risk of motor delay was greater than the risk of cognitive disability.	Weighted mean for infants assessed at 1 year of age: MDI: 90.3 (95% CL 88.9–91.6), PDI: 78.1 (95% CL 76.4–79.7). Lack of data at preschool and school age did not allow analysis and, therefore, results are not available for these age groups
Sterken et al. (2015) [27]	Intelligence, EF, attention, and memory	Individuals under the age of 24 years who underwent heart surgery or an interventional cardiac procedure vs. healthy control group	The CHD group had worse scores than healthy control children for all investigated neurocognitive functions. Intelligence and alertness were consistently affected, while memory was less affected. EF showed medium SMD with large heterogeneity.	Intelligence: SMD = −0.53 (95% CI −0.68 to −0.38); Alertness non-reaction time: SMD = −0.47 (95% CI −0.67 to −0.27); Alertness reaction time: SMD = 0.25 (95% CI 0.08 to 0.42); Verbal memory: SMD = −0.35 (95% CI −0.54 to −0.15); EF reaction time: SMD = 0.57; EF non-reaction time: SMD = −0.51 (95% CI −0.74 to −0.29)
Soares et al. (2023) [28]	MDI and PDI (BSID-II); Motor, Cognitive, and Language outcomes (BSID-III); IQ (WISC)	Children until 5 years of age with transposition of the great arteries surgically corrected during the neonatal period vs. normative data	For children from 1 to 3 years old, the mean MDI and PDI were within the average values although a significantly higher proportion of children scored more than 1 standard deviation below the mean in PDI, MDI, motor, and language composite scores compared to the general population. For children from 4 to 5 years, mean FIQ, VIQ, and PIQ did not differ significantly from the general population	Children at 1 year of age: BSID-II: mean PDI = 91.2 (95% CI 86.2–96.3); mean MDI was 96.2 (95% CI 88.5–104.0); BSID-III: 93.6 (95%CI 90.3–96.9); cognitive composite score 106.7 (95% CI 103.2–110.2). Children at 2 years of age: BSID-II: mean PDI 89.2 (95% CI 83.7–94.6); mean MDI was 90.8 (95% CI 82.8–98.8). BSID-III: mean motor composite score was 101.1 (95% CI 96.2–105.9); mean cognitive composite score was 100.8 (95% CI 92.9–108.7); mean language composite score was 94.1 (95% CI 90.0–98.2) Children at 3 years of age: BSID-II: PDI was 95.5 (95% CI 90.1–100.9); mean MDI was 95.3 [95% CI 92.1–98.6]. Children from 4 to 5 years old: FIQ 97.5 (95% CI 90.0–104.9); PIQ 92.9 (95% CI 89.7–96.2); VIQ 95.1 (95% CI 93.0–97.2)

Note. IQ—Intelligence Quotient; VIQ—Verbal Intelligence Quotient; PIQ—Performance Intelligence Quotient; FSIQ—Full-Scale Intelligence Quotient; EF—Executive function; CHD—Congenital heart disease; HLHS—Hypoplastic left heart syndrome; ASD—Atrial septal defects; VSD—Ventricular septal defects; S1N—Stage 1 Norwood Procedure; BSID—Bayley Scales of Infant Development; MDI—Mental developmental index; PDI—Psychomotor developmental index; PDMS—Peabody Developmental Motor Scale; WPPSI—Wechsler Preschool and Primary Scale of Intelligence; Griffiths MDS—Griffiths Mental Development Scale; WISC—Wechsler Intelligence Scale for Children; CI—Confidence interval; SMD—Standardized mean difference; g—Standardized weighted mean effect sizes.

## Data Availability

The original contributions presented in the study are included in the article; further inquiries can be directed to the corresponding author.

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
