# Peer review of "Cognitive Functioning in Children and Young People with Congenital Heart Disease: A Systematic Review of Meta-Analyses"

_healthcare, 2024, doi:10.3390/healthcare12242594_

Round 1
Reviewer 1 Report
Comments and Suggestions for Authors
In several places, comments are made implying that the studies included provide information about the time course of deficits suffered by children with congenital heart disease: e.g, “effects intensify with age” or “progressive decline in cognitive scores with increasing age.” These inferences are not warranted unless within-individual changes with age were documented in the studies, and the authors provide no evidence for this. In other words, it is not valid to draw this inference if all we know is that in two studies, children assessed at the older age did more poorly than children assessed at the younger age. There could be many reasons for this other than worsening performance over time within individuals
The authors should acknowledge that comparisons of studies done decades apart could be confounded by a survival bias—as noted, survival rates of children with more critical CHD have increased. Sicker children, who might be at greater risk of cognitive impairments, now survive. Such children would not have contributed data to studies conducted in an earlier era. This could also contribute to a finding that the performance of children with CHD has not improved dramatically over time.
The authors recommendation that Cardiology Centers include psychiatrists, psychologists, and behavioral pediatricians, and other professionals with expertise in neurodevelopment is important. They might mention the Cardiac Neurodevelopment Outcomes Collaborative (CNOC; https://cardiacneuro.org/), a relatively new organization of more than 50 hospitals and universities, that promotes attention to the issues covered in this review.
Author Response
In several places, comments are made implying that the studies included provide information about the time course of deficits suffered by children with congenital heart disease: e.g, “effects intensify with age” or “progressive decline in cognitive scores with increasing age.” These inferences are not warranted unless within-individual changes with age were documented in the studies, and the authors provide no evidence for this. In other words, it is not valid to draw this inference if all we know is that in two studies, children assessed at the older age did more poorly than children assessed at the younger age. There could be many reasons for this other than worsening performance over time within individuals
These comments are not based on individual empirical studies but on the meta-regressions reported in the respective meta-analysis by Siciliano et al. 2019. As described by Cochrane " Meta-regressions are similar in essence to simple regressions, in which an outcome variable is predicted according to the values of one or more explanatory variables. In meta-regression, the outcome variable is the effect estimate (for example, a mean difference, a risk difference, a log odds ratio or a log risk ratio). The explanatory variables are characteristics of studies that might influence the size of intervention effect. These are often called ‘potential effect modifiers’ or covariates." Hence, the meta-regression findings do not necessarily include causal data but infer from observational data. In the manuscript, it is clearly indicated that these inferences stem from the meta-regression findings so should be taken with some limitations.(see lines 304-307)
The authors should acknowledge that comparisons of studies done decades apart could be confounded by a survival bias—as noted, survival rates of children with more critical CHD have increased. Sicker children, who might be at greater risk of cognitive impairments, now survive. Such children would not have contributed data to studies conducted in an earlier era. This could also contribute to a finding that the performance of children with CHD has not improved dramatically over time.
This is now addressed in lines 425-427
The authors recommendation that Cardiology Centers include psychiatrists, psychologists, and behavioral pediatricians, and other professionals with expertise in neurodevelopment is important. They might mention the Cardiac Neurodevelopment Outcomes Collaborative (CNOC; https://cardiacneuro.org/), a relatively new organization of more than 50 hospitals and universities, that promotes attention to the issues covered in this review.
We thank the reviewer for this very useful resource. We have now added it in the mansucript lines 403-407.
Reviewer 2 Report
Comments and Suggestions for Authors
Thank you for submitting your article to this journal. I have some comments on the methodology of this article.
1. Why did you count people 18-24 as children?
2. The search date is more than one year ago and needs an update.
3. The search strategy for each data base should be added as a supplementary file.
4. Would you please report the number of included studies for each database?
5. While gathering the meta- analyses on a topic is valuable, would you please clarify what is the added value of this study to the current literature? What new insights did you bring to the literature that previously were not mentioned in the meta-analyses?
6. Some of the included studies in each meta-analysis are the same. How do you think it can affect your results?
Thank you.
Author Response
Why did you count people 18-24 as children?
Clarification and reference added in regard to this, see lines 85 – 88.
The search date is more than one year ago and needs an update.
The searches have been updated to November 27th, 2024
The search strategy for each data base should be added as a supplementary file.
We have added this information in appendix A at the end of the manuscript.
Would you please report the number of included studies for each database?
This is now detailed in lines 113-117 and on the PRISMA flowchart
While gathering the meta- analyses on a topic is valuable, would you please clarify what is the added value of this study to the current literature? What new insights did you bring to the literature that previously were not mentioned in the meta-analyses?
The unique contributions of this study to the existing body of knowledge have been further discussed in lines 374-386.
Some of the included studies in each meta-analysis are the same. How do you think it can affect your results?
This is now mentioned in the limitations line 420
Reviewer 3 Report
Comments and Suggestions for Authors
This is a well-written systematic review of meta-analyses investigating the potential association between CHD and cognitive functions.
I have a few comments:
1: the authors briefly presented the biological explanation of the association in the introduction section. I think this part needs to be further explained in the discussion section.
2: The included meta-analyses showed elevated I2, suggesting high heterogeneity across the included studies. The sources of this heterogeneity should be further explained in the limitation section. The author may consider presenting the results of the sensitivity analyses conducted in previous meta-analyses and clarify whether they could minimize this heterogeneity.
3: Lines 303 and 324: CHD was already spelled. The same goes for EF in lines 294 and 345 and HLHS in lines 310. Please revise all the abbreviations across the study to ensure they were spelled once in the first.
Author Response
1: the authors briefly presented the biological explanation of the association in the introduction section. I think this part needs to be further explained in the discussion section.
These have been addressed in the discussion section: lines 271-283 and, 345-360.
2: The included meta-analyses showed elevated I2, suggesting high heterogeneity across the included studies. The sources of this heterogeneity should be further explained in the limitation section. The author may consider presenting the results of the sensitivity analyses conducted in previous meta-analyses and clarify whether they could minimize this heterogeneity.
These are now adressed in the limitations section lines 372-385
3: Lines 303 and 324: CHD was already spelled. The same goes for EF in lines 294 and 345 and HLHS in lines 310. Please revise all the abbreviations across the study to ensure they were spelled once in the first.
These have now been amended
Round 2
Reviewer 2 Report
Comments and Suggestions for Authors
Thank you for the revisions. Please double check your search strategy for EMBASE. While you found only 11 citations in pubmed, there are more than 1200 citations in EMBASE.
Author Response
We would like to thank the reviewer for their time and effort in calling our attention to a discrepancy in the manuscript. We provide below a detailed response (in italic) and changes will be highlighted in yellow in the revised manuscript for ease of reading.
Reviewer comment:
“Please double check your search strategy for EMBASE. While you found only 11 citations in pubmed, there are more than 1200 citations in EMBASE."
We greatly appreciate the comment from the reviewer as there was a system error in the searches and EMBASE results had not been included. We have revised the searches with adequate parameters and did manage to retrieve a new article to include in the review. However, we did not retrieve anywhere near 1200 citations. We believe this might be due to a different use of the filter functions.
We have detailed the updated search in the manuscript, corrected the Prisma diagram, and where necessary adjusted the results reporting and discussion in the light of the new article.
We believe the manuscript conforms now with the request from the reviewer and we hope that the adjustments have been satisfactory.